# pH-Responsive Viscoelastic Fluids of a C_22_-Tailed Surfactant Induced by Trivalent Metal Ions

**DOI:** 10.3390/molecules28124621

**Published:** 2023-06-07

**Authors:** Zhi Xu, Shuai Yu, Rong Fu, Ji Wang, Yujun Feng

**Affiliations:** 1Polymer Research Institute, State Key Laboratory of Polymer Materials Engineering, Sichuan University, Chengdu 610065, Chinayushuai@stu.scu.edu.cn (S.Y.); 2Chengdu Institute of Organic Chemistry, Chinese Academy of Sciences, Chengdu 610041, China; 3West China School of Public Health, Sichuan University, Chengdu 610065, China; 4TianFu YongXing Laboratory, New Theory and Technology of CO2 Capture Research Center, Chengdu 610217, China

**Keywords:** viscoelastic fluids, wormlike micelles, ultra-long chain surfactants, metal-coordination, pH-responsiveness

## Abstract

pH-responsive viscoelastic fluids are often achieved by adding hydrotropes into surfactant solutions. However, the use of metal salts to prepare pH-responsive viscoelastic fluids has been less documented. Herein, a pH-responsive viscoelastic fluid was developed by blending an ultra-long-chain tertiary amine, *N*-erucamidopropyl-*N, N*-dimethylamine (UC_22_AMPM), with metal salts (i.e., AlCl_3_, CrCl_3_, and FeCl_3_). The effects of the surfactant/metal salt mixing ratio and the type of metal ions on the viscoelasticity and phase behavior of fluids were systematically examined by appearance observation and rheometry. To elucidate the role of metal ions, the rheological properties between AlCl_3_− and HCl−UC_22_AMPM systems were compared. Results showed the above metal salt evoked the low-viscosity UC_22_AMPM dispersions to form viscoelastic solutions. Similar to HCl, AlCl_3_ could also protonate the UC_22_AMPM into a cationic surfactant, forming wormlike micelles (WLMs). Notably, much stronger viscoelastic behavior was evidenced in the UC_22_AMPM−AlCl_3_ systems because the Al^3+^ as metal chelators coordinated with WLMs, promoting the increment of viscosity. By tuning the pH, the macroscopic appearance of the UC_22_AMPM−AlCl_3_ system switched between transparent solutions and milky dispersion, concomitant with a viscosity variation of one order of magnitude. Importantly, the UC_22_AMPM−AlCl_3_ systems showed a constant viscosity of 40 mPa·s at 80 °C and 170 s^−1^ for 120 min, indicative of good heat and shear resistances. The metal-containing viscoelastic fluids are expected to be good candidates for high-temperature reservoir hydraulic fracturing.

## 1. Introduction

Viscoelastic fluids, based on surfactants, are typically “living polymers”, with unique rheological properties, viz., viscoelasticity [1]. The peculiar viscoelasticity of the solution originates from the spontaneous assembly of the surfactants into wormlike micelles (WLMs) [1,2]. A striking advantage of viscoelastic surfactant fluids over regular water-soluble polymers is their reversibly shear-degradable characteristics because WLMs are connected by weak physical interactions that can continuously break and reform [3]. Among numerous viscoelastic fluids, viscoelastic fluids of C_22_-tailed surfactants are arguably the most attractive for the following reasons. (i) The viscoelastic fluids made of C_22_-tailed surfactants exhibit stronger viscoelasticity and better thermostability compared to their short-chain counterparts [4]. (ii) Unlike short-chain surfactants that are derived from crude oil-based products, C_22_-tailed surfactants are environmentally benign and sustainable because their feedstocks are natural, renewable raw materials, such as vegetable oil [5,6,7]. Owing to their fascinating rheological behavior, C_22_-tailed surfactant viscoelastic fluids have shown great potential in a wide range of areas, such as clean fracturing fluids [8], drag reduction [9], and personal care products [10].

In some cases, the viscoelasticity of the fluid needs to be tailored to the demand [11]. Taking hydraulic fracturing as an example [12,13], the high viscosity of fracturing fluids is typically needed to transport sand particles in the construction process. After fracturing, fracturing fluids with low viscosity are preferably considered, as they can flow back smoothly, thus minimizing fracture blockage. The general strategy to achieve such a purpose is to use internal breakers to reduce the viscoelasticity of flow-back fracturing fluid [14]. Unfortunately, these external additives would not only change the composition of the viscoelastic fluids, but they would also prevent their recovery and reuse. Therefore, much attention has been shifted to constructing smart viscoelastic fluids, which can be switched reversibly between water-like and viscoelastic states [15]. Of these stimulus-responsive systems, pH-responsive viscoelastic solutions possess the advantages of simple preparation, controllability, easy operation, and being cost-effective [16]. A classical methodology for fabricating pH-switchable viscoelastic solutions is to introduce organic acid to the surfactant system [17]. For instance, Huang and co-workers [18] employed cetyltrimethylammonium bromide and potassium phthalic acid to prepare a pH-responsive viscoelastic solution that undergoes a fully reversible, repeatable “sol–gel” transition within a narrow pH region from 3.90 to 5.35. Likewise, our group [17] developed a pH-switchable viscoelastic solution by mixing a long-chain tertiary amine, *N*-erucamidopropyl-*N, N*-dimethylamine (UC_22_AMPM), and maleic acid in a 2:1 molar ratio. It was found that the solution exhibited a reversible Newtonian-to-viscoelastic transition with an increase in zero-shear viscosity (*η*_0_) by five orders of magnitude as the pH increased from 6.20 to 7.29. Correspondingly, the micellar aggregate showed a structural transformation from spheres to WLMs. Nevertheless, it has been recognized that these organic acids, such as maleic acid, experienced decomposition or polymerization [16] at elevated temperatures, impairing the pH responsiveness and viscoelasticity of fluids. Obviously, such systems are not available for high-temperature reservoirs. In this context, it is desirable and beneficial to develop novel pH-responsive viscoelastic fluids with good temperature-resistance properties.

During past decades, the development of metal ion–ligand coordination bonds has also been a common approach to constructing pH-responsive materials. The mechanism behind this is that such a bond would break under acidic conditions, causing changes in the macroscopic properties of the material as H^+^ replaces the metal ion bound to the ligand [19,20]. At alkaline conditions, the metal ion–ligand coordination bonds would reform [21]. For example, Che et al. [20] developed a novel pH-responsive mesoporous silica drug delivery system by incorporating drug molecules into the mesopores silica using coordination bonding in a “host–metal–guest” architecture. Under weakly acidic conditions (pH 5.0-6.0), the encapsulated drugs were readily released in response to a reduction in pH due to a breakdown of coordination bonds. Given the excellent thermal stability of metal ions, a promising way to build a novel pH-responsive viscoelastic fluid, tolerating high temperatures, is to introduce metal ions that can coordinate with surfactant ligands.

To realize the abovementioned idea, a pH-responsive viscoelastic fluid was proposed by simple complexation of the ultra-long-chain tertiary amines UC_22_AMPM (Figure 1) and AlCl_3_. The chosen UC_22_AMPM is capable of providing a pair of electrons that can be deemed a Lewis base. AlCl_3_ was selected because it is a typical hard Lewis acid that coordinates hard basic centers, such as nitrogen atoms in amine groups, and then it forms coordination bonding. The phase behavior and rheological properties of the UC_22_AMPM–AlCl_3_ system were meticulously examined and compared with the UC_22_AMPM–HCl system to shed light on the role of AlCl_3_ in UC_22_AMPM. Meanwhile, the pH-responsiveness of the UC_22_AMPM–AlCl_3_ system was illustrated by macroscopic appearance observation and rheological measurement at different pH values. Then, cryo-transmission electron microscopy (cryo-TEM) and proton nuclear magnetic resonance spectroscopy (^1^H NMR) were employed to unravel the intrinsic mechanism involved in the pH-responsiveness of the UC_22_AMPM–AlCl_3_ mixed system. Finally, the properties of the UC_22_AMPM–AlCl_3_ viscoelastic solution as a fracturing fluid were evaluated in terms of temperature tolerance and shear tolerance.

## 2. Results and Discussion

### 2.1. Phase Behavior of UC_22_AMPM–AlCl_3_ System

The macroscopic appearance photos of the UC_22_AMPM–AlCl_3_ system at different *α* (molar ratio *=* UC_22_AMPM:AlCl_3_) under *C*_UC22AMPM_ = 50 mM are shown in Figure 1. One can find that, with increasing AlCl_3_ concentration, the UC_22_AMPM dispersion transformed from an opalescent dispersion to a transparent solution, indicative of the increment in water solubility of the UC_22_AMPM molecules. Remarkably, at *α* = 1:9, the mixed system exhibited phase separation.

To unveil the underlying reasons for the phase behavior variation, pH and ^1^H NMR measurements were carried out. It can be seen from Figure 2A that the pH of the sample dramatically decreased from 8.17 to 3.73 as the AlCl_3_ concentration rose from 0 to 50 mM. The reducing pH was associated with the hydrolysis of AlCl_3_ in water, yielding substantial amounts of H^+^ in the solution. There is also no appreciable change in the pH values with further increase in AlCl_3_ concentration, indicating that the AlCl_3_ hydrolysis has reached equilibrium. Figure 2B compares the ^1^H NMR spectra of UC_22_AMPM to those of a sample in which AlCl_3_ (*α =* 1:3) was added. In comparison to neat UC_22_AMPM, the chemical shifts of protons neighboring amine groups (peaks a, b, and c) of UC_22_AMPM molecules in the presence of 150 mM AlCl_3_ were shifted from 1.74, 2.47, and 2.30 ppm to 1.96, 3.15, and 2.91 ppm, respectively, manifesting the protonation of the tertiary amine group. Combined with the results of pH and ^1^H NMR, it was suggested that the hydrolysis of AlCl_3_ in water yields substantial amounts of H^+^, leading to the protonation of UC_22_AMPM. Consequently, UC_22_AMPM behaves like a cationic surfactant, exhibiting good water solubility.

As for phase separation, it may be due to the fact that the excessive AlCl_3_ salted the UC_22_AMPM compounds out by dehydration, lowering their water solubility.

### 2.2. Rheological Properties of the UC_22_AMPM–AlCl_3_ Mixed System

To further shed light on the effect of AlCl_3_ on the UC_22_AMPM solution, the rheological properties of UC_22_AMPM–AlCl_3_ with different *α* were studied. As depicted in Figure 3A, in the cases of *α* = 1:0, 9:1, and 6:1, the viscosities (*η*) of the solutions were close to that of water, and they were independent of shear rate. That proved that these fluids were typical Newtonian fluids, implying the existence of only spherical micelles in the above solutions. Meanwhile, the UC_22_AMPM–AlCl_3_ samples at *α* = 3:1, 1:1, 1:3, and 1:6 presented a Newtonian plateau and shear-shinning behavior at low- and high-shear rate regions, respectively, reflecting the formation of WLMs. To gain insight into the rheological properties of the UC_22_AMPM–AlCl_3_ solution, we performed dynamic rheological scans at 25 °C on the UC_22_AMM–AlCl_3_ sample at *α* = 1:3. As indicated in Figure 3B, it can be seen that the sample showed a gel-like response, i.e., the *G*′ exceeds *G*″ over the frequency range. Similar rheology behavior has been extensively observed in the WLMs of C_22_-tailed surfactants by both Raghavan [22,23] and Feng [22], mirroring the extremely long relaxation times of WLMs of UC_22_AMPM–AlCl_3_ complexes.

For comparison, we also investigated the steady rheology of the UC_22_AMPM systems in the presence of HCl. As is exhibited in Figure 4A, at pHs of 7.14 and 6.88, the UC_22_AMPM–HCl mixtures behaved as water-like Newtonian fluids with a constant *η* of ~3 mPa·s. At pH below 6.88, the mixtures displayed Newtonian plateau and shear-thinning behavior. The similarity in rheological behavior of the above two systems signified that AlCl_3_ plays a comparable role to HCl in UC_22_AMPM. That is, UC_22_AMPM can be protonated by adding either HCl or AlCl_3_, converting the cationic state and self-assembling into WLMs.

To further compare the effects of AlCl_3_ and HCl on the rheological properties of UC_22_AMPM solution, the zero-shear viscosity (*η*_0_) of the UC_22_AMPM solution was plotted as a function of AlCl_3_ concentration or pH (Figure 4B). It was found that *η*_0_ increased by five orders of magnitude when the AlCl_3_ concentration increased from 15 to 30 mM. Following an increase in AlCl_3_ concentration above 30 mM, the *η*_0_ almost remained unchanged at 10^5^ mPa·s. In contrast, the *η*_0_ of UC_22_AMPM solution increased by three orders of magnitude as HCl increased. Apparently, AlCl_3_ is more efficient in thickening the UC_22_AMPM dispersion as compared to HCl. A plausible explanation could be that the tertiary amine of UC_22_AMPM can coordinate with trivalent metal ions, i.e., Al^3+^, leading to cross-linking of the wormlike micellar chains, enhancing the solution viscosity [24,25,26]. Similar results were reported by Hao et al. [25], who found the presence of trivalent metal ions (Fe^3+^ and Al^3+^) increased the viscosity of WLMs of tetradecyldimethylamine oxide and amphiphilic short peptides by 10- and 25-fold, respectively.

### 2.3. Effect of Other Trivalent Metal Salts on UC_22_AMPM

It is well known that properties of the coordinated ion, such as ionic radius and crystal field stabilization energy, profoundly affect the strength of ion–ligand coordination bonds and the solution viscoelasticity [24,27]. To clarify the influence of these different trivalent metal ions on the rheological behavior of UC_22_AMPM, two trivalent metal salts (FeCl_3_ and CrCl_3_) were separately added to 50 mM UC_22_AMPM dispersion, and the resulting mixed samples were examined by a combination of rheometer and visual observations.

As depicted in Figure 5A,B, both mixtures underwent a transformation from orange or white dispersions to red or black translucent solutions as the metal salt concentration (*C*_MCl3_) increased. The different solution colors originated from the nature of Fe^3+^ or Cr^3+^ ions [28]. Steady rheology spectra of the UC_22_AMPM–FeCl_3_ and UC_22_AMPM–CrCl_3_ systems are shown in Figure 5C,D, and it was demonstrated that both systems also exhibited similar transitions from a water-like, low-viscosity fluid to a translucent viscoelastic fluid with increasing *β* (molar ratio *=* UC_22_AMPM:FeCl_3_) and *χ* (molar ratio *=* UC_22_AMPM:CrCl_3_). This finding demonstrated that the trivalent metal ions could induce UC_22_AMPM dispersion to form viscoelasticity fluid, regardless of the type of trivalent metal ions.

To explore the efficiency of viscosity enhancement, the effect of the above two cases of *C*_MCl3_ on *η*_0_ of the UC_22_AMPM solution was further investigated. From the results shown in Figure 6A, it was clear that the *η*_0_ of both cases first remained constant when the *C*_MCl3_ was lower than 10 mM, and then it sharply rose in the *C*_MCl3_ range of 10–25 mM, and, finally, it reached a viscosity plateau once the *C*_MCl3_ exceeded 25 mM. Note that the above two solutions achieved a *η*_0_ of 10^5^ mPa·s within the studied salinity scope, suggesting the identical thickening capability of both metal salts (i.e., FeCl_3_ and CrCl_3_). Interestingly, a relatively lower amount of FeCl_3_ (20 mM) was required to reach such a high *η*_0_ compared to CrCl_3_ (24 mM), meaning that the efficiency of viscosity enhancement of Fe^3+^ is superior to that of Cr^3+^. In Figure 6B, the pH for the above two cases is compared as a function of *C*_MCl3_. Overall, the pH of the UC_22_AMPM–metal salt mixtures showed a decreasing trend within the studied salinity scope, which can also be interpreted in relation to the hydrolysis of metal salts. It is noteworthy that the pH of the UC_22_AMPM–CrCl_3_ sample was higher than that of the UC_22_AMPM–FeCl_3_ sample under identical *C*_MCl3_, signifying that FeCl_3_ is more adequately hydrolyzed than CrCl_3_. Therefore, we attributed the preferable viscosity-enhancing efficiency of Fe^3+^ to the more adequate hydrolysis of FeCl_3_.

### 2.4. pH Responsiveness of the UC_22_AMPM-AlCl_3_ Mixed System

As stated, the metal ion–ligand coordination bonds are sensitive to external pH variations because both metal ions and H^+^ ions compete to combine with the ligand. Therefore, it is expected that the UC_22_AMPM–AlCl_3_ blends would exhibit tunable viscoelasticity by altering the pH of the solution. To prove this concept, the UC_22_AMPM–AlCl_3_ mixed system at *α* = 1:3 was chosen as a representative sample and characterized concerning its rheological behavior at different pH values. As observed in Figure 7A, the *η*_0_ of the UC_22_AMPM–AlCl_3_ mixed system increased slightly, then rapidly reduced, and finally remained intact as pH increased. From the insets of Figure 7A, the sample was the bluish transparent solution and milky turbid fluid at the examined pH levels (pH 3.75 and 7.05), respectively, which were related to the protonation and deprotonation of UC_22_AMPM. To be specific, the UC_22_AMPM was protonated and thus it easily dissolved in water in acidic conditions. Under alkaline conditions, UC_22_AMPM preferred non-ionic species and showed poor solubility due to its extremely long hydrophobic tail, resulting in the formation of turbid solutions. Impressively, unlike regular pH-responsive systems, which switch between gel-like and water-like states [29,30], the *η*_0_ of UC_22_AMPM–AlCl_3_ varied by merely one order of magnitude at the examined pH scope (1.30–8.78). According to our previous studies [29], it was demonstrated that the nonionic UC_22_AMPM would self-assemble vesicles in basic conditions. Here, we proposed that the vesicles could also coordinate with Al^3+^ ions to form metal ions–vesicle complexes, blocking a substantial solution viscosity reduction.

Figure 7B describes the curve of pH-stimulated reversibility for the UC_22_AMPM–AlCl_3_ mixed system. As the cycle number increased, the *η*_0_ of the blends diminished slightly at both pH 7.05 and 3.75, reflecting their poor switchability. We attributed this to the formation of by-products during the repeated addition of acids and bases, which would deteriorate the viscoelasticity of fluids.

To reveal the underlying reasons for the variation of macroscopic properties, the morphology of the UC_22_AMPM–AlCl_3_ mixed system with *α* = 1:3 at different pH values was directly visualized by cryo-TEM. As shown in Figure 7C, high-density, long, and flexible WLMs were observed in the UC_22_AMPM–AlCl_3_ mixed system at pH 3.75, and it is difficult to identify where they begin and end. These WLMs overlapped and entangled with each other into three-dimensional network structures, accounting for the gel-like response and high *η*_0_ of this sample. On the contrary, when increasing pH to 7.05, only spherical vesicles are observed (Figure 7D), consistent with our previous inference (Figure 3A and Figure 7A).

Based on the above results, we proposed the following mechanism to account for the effect of metal salts on the UC_22_AMPM (Figure 2). The addition of metal salts (AlCl_3_) was first hydrolyzed to generate abundant trivalent metal ions (Al^3+^) and H^+^ (Equations (1) and (2)). In this scenario, the resulting H^+^ first protonated the UC_22_AMPM molecules to their cationic form (Equation (3)), inducing the formation of WLMs. Entanglement of these WLMs into a transient network imparts high *η* to solutions. More importantly, Al^3+^ was tightly associated with the headgroups of UC_22_AMPM to form metal–WLM ligand-coordinated systems by coordinating interaction, further enhancing the *η* of the solution [31]. These reactions could be expressed as follows:AlCl_3_ + 3H_2_O ⇋ Al(OH)_3_ + 3H^+^
(1)
Al(OH)_3_ ⇋ Al^3+^ + 3OH^−^
(2)
UC_22_AMPM + H^+^ → UC_22_AMPM·H^+^
(3)

Upon decreasing pHs (by adding HCl solution), the UC_22_AMPM molecules still maintained protonation states, and thus the corresponding wormlike micellar structure remained unchanged. Meanwhile, the reaction of H^+^ with Al(OH)_3_ yields a larger amount of Al^3+^ (Equation (4)). There is no doubt that the increment in the amount of Al^3+^ further boosted the formation of metal–WLM ligand coordination, enhancing the entanglement density of the WLMs network and, thereby, leading to a remarkable enhancement of the solution *η*.
Al(OH)_3_ + 3H^+^ → Al^3+^ + 3H_2_O (4)

Conversely, the UC_22_AMPM molecules converted into nonionized forms upon increasing pH (addition of NaOH into the aqueous solution), leading to a transformation of the aggregate structure from wormlike to the vesicle. More importantly, the OH^−^ induced Al(OH)_3_ to produce a larger amount of AlO_2_^−^ (Equation (5)), reducing the amount of Al^3+^. As a result, the metal–WLM ligand–coordinated and entangled WLM networks were broken, rendering a decrease in viscosity.
Al(OH)_3_ + OH^−^ → AlO_2_^−^ + 2H_2_O (5)

It is worth noting that the *η*_0_ of UC_22_AMPM−AlCl_3_ mixtures was reduced by only one order of magnitude as pH increased from 3.75 to 7.05, which was due to the presence of metal–vesicle coordinated complexes.

### 2.5. Temperature Tolerance and Shear Tolerance of the UC_22_AMPM−AlCl_3_ Mixed System

At present, viscosity loss is the most prominent defect in clean fracturing fluids at high temperatures and high shear rates. Therefore, temperature- and shear-resistant properties have been considered the major criteria for evaluating the applicability of viscoelastic fluids in hydraulic fracturing [15].

To evaluate the potential of UC_22_AMPM–AlCl_3_ mixed systems as clean fracturing fluids, high temperature and high shear measurements were performed under simulated fracturing conditions. As shown in Figure 8A, the *η* at 170 s^−1^ of UC_22_AMPM–AlCl_3_ mixed system at *α* = 1:1 gradually decreased and then maintained a constant viscosity of ~60 mPa·s with increasing temperature from 25 to 65 °C. Over the temperature range from 65 to 95 °C, the UC_22_AMPM–AlCl_3_ mixed system further declined to ~40 mPa·s, reflecting its good heat tolerance. Again, this can be interpreted by the fact that Al^3+^ ions are very tightly bound to the UC_22_AMPM by coordination interaction, which greatly promotes the thermal stability of WLMs.

Moreover, at temperatures of 80 and 60 °C, the UC_22_AMPM–AlCl_3_ mixed system can keep a stable viscosity around 40 mPa s and 60 mPa s for 120 min (Figure 8B), respectively, indicating the good shear tolerance of the UC_22_AMPM–AlCl_3_ mixed system. More importantly, it is also higher than the viscosity requirements (>25 mPa s) for clean fracturing fluid [32]. Therefore, it was believed that the UC_22_AMPM–AlCl_3_ mixed system can satisfy the demand for clean hydraulic fracturing in the vast majority of oil fields, from middle-low-temperature reservoirs to high-temperature reservoirs.

## 3. Materials and Methods

### 3.1. Materials

UC_22_AMPM was synthesized, according to our previously-reported procedure [33], and confirmed by proton nuclear magnetic resonance spectroscopy (^1^H NMR, Figure 2B). Hydrogen chloride (HCl, 34 vol.%), sodium chloride (NaCl, 99%, GC), aluminium chloride (AlCl_3_, 99%, GC), ferric chloride (FeCl_3_, 99%, GC), and chromium trichloride (CrCl_3_, 99%, GC) were purchased from Chengdu Kelong Chemical Factory Co., Ltd. (Chengdu, China) and were used as received. CD_3_OD and D_2_O (both with 98% deuterium content) used for NMR analysis were obtained from Sigma-Aldrich (Shanghai, China). Deionized water with a resistivity of 18.25 MΩ·cm used throughout this study was prepared from a quartz water purification system (UPH-I-10T, Chengdu Ultra-pure Technology Co., Ltd., Chengdu, China).

### 3.2. Preparation of UC_22_AMPM–Metal Salts Mixtures

A stock dispersion with 50 mM UC_22_AMPM was prepared by adding designed amounts of power-like samples and deionized water to a sealed Schott-Duran bottle equipped with a magnetic bar inside, followed by gentle agitation at 60 °C, yielding a low-viscosity emulsion-like dispersion. Then, the desired amount of inorganic metal salts was added into a 50 mM UC_22_AMPM dispersion, followed by mechanical agitation for 12 h. The samples were left at 25 °C for at least 24 h prior to measurements. The dilute NaOH and HCl solutions were employed to adjust the pH of the UC_22_AMPM–metal salt mixtures; the pH was determined by a Sartorius basic pH meter PB-10 (±0.01).

### 3.3. Rheology Measurement

Rheological properties of mixture solutions were performed on a Physica MCR 302 (Anton Paar, Graz, Austria) rotational rheometer equipped with a concentric cylinder geometry CC27 (ISO3219). Samples were equilibrated at the testing temperature for no less than 20 min prior to the experiments. During steady–shear tests, the $\dot{\gamma }$ and test time (t) parameters were varied logarithmically from 1 × 10^−4^ to 1.5 × 10^3^ s^−1^ and 1 × 10^4^ to 1 s, respectively, according to the relationship $\dot{\gamma }$ × t ≥ 1. The extrapolation of *η* to zero-shear rate in the steady-shear measurement yields the zero-shear viscosity, *η*_0_.

For temperature sweep measurements, solution viscosity was recorded at a shear rate of 170 s^−1^ at various temperatures, ranging from 25 to 95 °C. The heating rate was fixed at 1 °C/min to ensure that the sample was equilibrated. Sufficient time was allowed before data collection at each temperature to ensure the viscosities reached their steady values.

The oscillatory measurements were conducted in the fixed stress (linear viscoelastic region), as determined from prior dynamic stress sweep measurements. The frequency varied from 0.01–100 rad·s^−1^. All measurements were carried out in stress-controlled mode, and Canon standard oil was used to calibrate the instrument before measurements. The temperature was controlled at 25 ± 0.1 °C using a Peltier device, and a solvent trap was used to minimize water evaporation during the measurements. For all experiments, flow curves were registered in a stress-controlled mode, and the data were acquired by the software Rheoplus^TM^.

### 3.4. Micellar Structure Observation

The specimens of the UC_22_AMPM–AlCl_3_ mixed solution were prepared in a controlled environment vitrification system. The temperature of the chamber was maintained at about 25 °C, and the relative humidity was maintained close to saturation to prevent evaporation during preparation. Typically, 5 μL of sample solution was deposited on a copper grid and gently blotted with a piece of filter paper to obtain a thin liquid film (20–400 nm) on the grid. Next, the grid was plunged rapidly into liquid ethane (−183 °C) and transferred into liquid nitrogen (<−160 °C) for storage. Finally, the vitrified specimen was transferred into a FEI Talos F200C using a Gatan 626 cryo-holder and observed at an acceleration voltage of 200 KV and a temperature of −170 °C. The images were recorded digitally with a charge-coupled device camera under low-dose conditions with an under-focus of approximately 3 mm.

### 3.5. Phase Behavior Observation

Different amounts of melt salts were added to aliquots of UC_22_AMPM solution (50 mM) to give a range of final melt salt concentrations (*C*_s_ = 5.5 − 450 mM). The resulting colloidal dispersions were kept in a sealed glass vial at 25 °C for equilibration. Phase behavior was recorded by visual observation, following the previously reported procedure [34].

### 3.6. ^1^H NMR Experiments

An amount of 4.0 mg of sample, including the neat UC_22_AMPM and the mixtures of UC_22_AMPM and AlCl_3_, was dissolved in 0.6 mL of CD_3_OD and D_2_O (*V*/*V* = 5:1) mixed solvent. ^1^H NMR spectra were registered in a Bruker AC 400 spectrometer (Bruker Instruments, Mannheim, Germany) at a proton resonance frequency of 400 MHz. Chemical shifts were reported on the *δ* (ppm) scale. The accuracy of the chemical shift reading was ± 0.01 ppm.

## 4. Conclusions

In this work, we developed a pH-responsive viscoelastic fluid by complexing a C_22_-tailed surfactant, UC_22_AMPM, with AlCl_3_. For molar ratios of UC_22_AMPM to AlCl_3_
*=* 3:1, 1:1, 1:3, and 1:6, the UC_22_AMPM dispersion could form a transparent viscoelastic fluid, similar to that of the UC_22_AMPM–HCl mixed system. Due to the metal–ligand coordination between the Al^3+^ and N atoms of UC_22_AMPM, the viscosity of the UC_22_AMPM–AlCl_3_ mixed system was higher than that of the UC_22_AMPM–HCl system under identical pH values. By tuning the pH of the solution, the macroscopic behavior of the UC_22_AMPM–AlCl_3_ mixed system switched between milky dispersion and a clear viscoelastic solution, accompanied by changes in viscosity. The pH responsiveness was caused by the morphology transition between WLMs and spherical vesicles at different pHs. The temperature and shear resistance test revealed the UC_22_AMPM–AlCl_3_ system attained a viscosity of 40 mPa·s at 80 °C and 170 s^−1^ for 120 min, featuring good thermal and shear resistance. These benefits indicated that such viscoelastic fluids can be a promising clean fracturing fluid for the exploration of high-temperature reservoirs. In summary, this study successfully constructed a pH-responsive viscoelastic fluid by introducing trivalent metal ions in a C_22_-tailed surfactant and revealed the role of trivalent metal ions in viscoelastic fluids, enriching the methodology for the preparation of pH-responsive viscoelastic fluids.

## Data Availability

The data presented in this study are available upon request from the corresponding author.

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
