# Peer review of "pH-Responsive Viscoelastic Fluids of a C22-Tailed Surfactant Induced by Trivalent Metal Ions"

_molecules, 2023, doi:10.3390/molecules28124621_

Round 1
Reviewer 1 Report
Please see attached doc.

It needs some editing.
Reviewer 2 Report
I would request the authors to revise the manuscript so that the articles are attractive to the readers.
Materials and methods should be clearly stated, so that any one can reproduce the experiment to some extent.
Figure 1 and 2 are quite small and hard to see, which is discouraging.
Results and discussion is satisfactory, although it can improved.
The English needs to be brushed up especially after the introduction. There were several areas that I could not understand clearly because of this. There are typos as well.
Reviewer 3 Report
In the manuscript entitled "pH-Responsive Viscoelastic Fluids of a C22-tailed Surfactant Induced by Trivalent Metal Ions" the authors have developed the pH responsive viscoelastic fluid with the complexation of C22 tailed surfactant and trivalent metal ion like AlCl3. The proposed viscoelastic fluid forms WLMs and spherical vesicles at different pH and have good thermo as well as shear resistance properties. The manuscript is nice and well written. I will suggest to accept it in a current format.
Author Response
Response: We sincerely thank the referee for his/her positive comments and encouragement.